# Synthesis, Characterization, and Biological Activity of a Novel Series of Benzo[4,5]imidazo[2,1-*b*]thiazole Derivatives as Potential Epidermal Growth Factor Receptor Inhibitors

**DOI:** 10.3390/molecules24040682

**Published:** 2019-02-14

**Authors:** Xinshan Deng, Xiaoyu Tan, Tiantian An, Qingqing Ma, Zhe Jin, Ce Wang, Qingguo Meng, Chun Hu

**Affiliations:** 1Key Laboratory of Structure-Based Drug Design & Discovery, Ministry of Education, Shenyang Pharmaceutical University, Shenyang 110016, China; xinshandeng@126.com (X.D.); tanxiaoyuaikaoyan@163.com (X.T.); antiantiansyphu@163.com (T.A.); maqingqing01@163.com (Q.M.); wangce0412@163.com (C.W.); 2Department of Pharmacy, Yantai University, Yantai 264005, China; qinggmeng@163.com

**Keywords:** EGFR inhibitors, heterocycle, synthesis, molecular docking, antitumor activity

## Abstract

Based on the analysis of epidermal growth factor receptor (EGFR) complexes with gefitinib with molecular docking, the scaffold-hopping strategy, combination of the active substructures, and structural optimization of EGFR inhibitors, a novel series of benzo[4,5]imidazo[2,1-*b*]thiazole derivatives was designed, synthesized, and evaluated for antitumor activity in human cancer cell lines and cellular toxicity against human normal cell lines using the 3-(4,5-dimethylthiazol-2-yl)-2,5-diphenyltetrazolium bromide (MTT) colorimetric assay and EGFR inhibitory activities in vitro. Some target compounds such as 2-(benzo[4,5]imidazo[2,1-*b*]thiazol-3-yl)-N-(2-hydroxyphenyl)acetamide (**D04**) and 2-(benzo[4,5]imidazo[2,1-*b*]thiazol-3-yl)-N-(naphthalen-1-yl)acetamide (**D08**) have shown significant antitumor activity against the EGFR high-expressed human cell line HeLa. All the target compounds showed hardly any antitumor activity against the EGFR low-expressed human cell line HepG2, and nearly no cellular toxicity against the human normal cell lines HL7702 and human umbilical vein endothelial cell lines (HUVEC). The inhibitory activities against EGFR kinase in vitro of the three target compounds were greatly consistent with the anti-proliferative activities. The preliminary structure–activity relationships of the target compounds were summarized. Conclusively, the novel benzo[4,5]imidazo[2,1-*b*]thiazole derivatives as novel potential EGFR inhibitors may be used as the potential lead compounds for the development of antitumor agents.

## 1. Introduction

Presently, many populations are affected by cancer. In 2012, 14.2 million new cancer cases and 8.2 million deaths occurred, and by 2030, this statistic is reported to increase to about 19 million [1].

To find an effective way to prevent cancer from developing and stop patients from deteriorating, over the past decade, a novel series of selective chemotherapeutic drugs such as imatinib, gefitinib, erlotinib, and afatinib has been launched worldwide for the treatment of cancer (Figure 1). The action targets of these innovative drugs are mainly directed to impact the signal transduction pathway of the epidermal growth factor receptor (EGFR, Figure 2) [2].

The over-expression and/or mutation of EGFR plays an important role in the growth of cancer including cell proliferation, anti-apoptosis, metastasis, and angiogenesis [3]. The EGFR has been reported as an attractive target for cancer therapy [4,5,6]. Over recent years, many scientists have discovered the mechanism of the EGFR that the binding of specific ligands such as epidermal growth factor (EGF) and transforming growth factor α (TGFα) to the extracellular domain of EGFR induces the dimerization, activation of intrinsic kinase activity, and subsequent autophosphorylation of EGFR at multiple residues in the cytoplasmic region, while the downstream signaling proteins including phosphoinositide 3-kinase (PI3-K), Akt, mammalian target of rapamycin (mTOR), Ras, Raf, MEK, and extracellular signal-regulated kinase (ERK) stimulate several signal transduction cascades, leading to DNA synthesis and cell proliferation (Figure 2) [7,8,9,10]. As it can obviously influence the proliferation of human cancer cells, and even result in the apoptosis of human cancer cells, the EGFR has become a very luminous target for various antitumor candidates [11].

Gefitinib, as a selective small-molecule EGFR tyrosine kinase inhibitor indicated for the treatment of adults with locally advanced or metastatic non-small cell lung cancer (NSCLC) with activating mutations of EGFR tyrosine kinase, was approved by the USA Food and Drug Administration (FDA) in 2003 [11]. In this study, to discover a novel EGFR inhibitor, gefitinib was used as the lead molecule. According to the scaffold-hopping strategy of drug design [12], the scaffold of the lead molecule was changed to benzo[4,5]imidazo[2,1-*b*]thiazole, a novel heterocycle core. As the two side chains of the quinazoline ring in gefitinib serve to improve physical properties and confer a favorable pharmacokinetic profile in animals and humans [13], based on the efficacy of the target compounds that should be preferentially considered, the side chains in the core scaffold were tentatively ignored. The pharmacodynamic aniline group in gefitinib was retained in the target compounds with a –CH_2_CO– linker (Figure 3). Moreover, to gain an understanding of the interactions between these benzo[4,5]imidazo[2,1-*b*]thiazole derivatives and EGFR, hereinafter docking studies are also presented. Herein, we report on the synthesis, characterization, and biological activity against cancer and normal human cell lines of a novel series of benzo[4,5]imidazo[2,1-*b*]thiazole derivatives as EGFR inhibitors.

## 2. Results and Discussion

### 2.1. Virtual Screening

In general, the molecular docking technique can help to understand the binding mode of the target compounds with EGFR, rationalize the molecular level interactions between the bio-target and ligand, and play an important role in new drug discovery.

In this study, gefitinib was chosen as the lead molecule, which was transformed to become the target compounds. The crystallographic structure of the EGFR-gefitinib complex [14], which was encoded as 4I22 in the Protein Data Bank (PDB) of Research Collaboratory for Structural Bioinformatics (RCSB), was used for the docking model.

Molecular docking studies between the EGFR and the target compounds with gefitinib were performed with Molegro Virtual Docker 2010 (v4.1) (Molegro ApS, Aarhus, Denmark). The docking algorithm was set at 1500 maximum iterations with a simple evolution population size of 50 and a minimum of 10 runs. Its original inhibitor (gefitinib) was deleted and all the water molecules were removed, adding a further degree of freedom in terms of variability in the docking results. Schematic diagrams of the interactions between the EGFR and the small molecules were analyzed by the Discovery Studio 2016 client and the PyMOL. All parameters of the binding energy and the H-bond energy of the target compounds were revealed to be as good as the lead one (Table 1).

Based on the docking results, a significant hydrogen bond was formed with MET793 for gefitinib, and the other EGFR residues that interacted with gefitinib were LEU718, VAL726, ALA743, LYS745, LEU788, MET790, GLN791, MET793, PRO794, LEU844, and THR854 (Figure 4). Surprisingly, there was an equal hydrogen bond that formed with MET793, and the same other EGFR residues interacted excluding GLN791 with EGFR for the target compounds D04 and D08 (Figure 5). Therefore, the docking results indicate that the target compounds would become potent EGFR inhibitors.

### 2.2. Chemistry

As shown in Scheme 1, all the target compounds were rooted from benzene-1,2-diamine (**1**). The 1*H*-benzo[*d*]imidazole-2-thiol (**2**) was prepared by **1** reacted with CS_2_ and Na_2_CO_3_, and then easily converted to the intermediate **3** with ethyl 4-chloro-3-oxobutanoate. Subsequently, the intermediate **4** was prepared by cyclization of the intermediate **3** in the presence of concentrated sulfuric acid at 50 °C, and then the intermediate **4** was directly hydrolyzed and acidified to yield the intermediate **5**. Finally, the reaction between the intermediate **5** and aniline or aniline derivative resulted in the target compounds **D01** to **D18**.

The chemical structures of the target compounds and intermediates were characterized by infrared (IR) spectra, ^1^H-NMR, ^13^C-NMR, ESI-MS, and high-resolution mass spectrometry (HRMS), and no target compounds were reported in the literatures.

### 2.3. Biological Evaluation

#### 2.3.1. In Vitro Anti-Proliferative Activity

The inhibitory activities of all target compounds against the cervical cancer cell line HeLa, human liver cancer cell line HepG2, human liver cell line HL7702, and human umbilical vein endothelial cell line HUVEC were determined by the 3-(4,5-dimethylthiazol-2-yl)-2,5-diphenyltetrazolium bromide (MTT) assay. As known, the EGFR level of the HeLa cells was highly expressed [15,16], while that of the HepG2 cells was low [17], and the HL7702 cells and HUVEC cells were human normal cells. The results expressed as IC_50_ values are summarized in Table 2.

As shown in Table 2, hardly any the target compounds exhibited a significant effect on HepG2, HL7702m, and HUVEC cells (IC_50_ > 100 µM for HepG2, IC_50_ > 200 μM for HL7702, IC_50_ > 100 μM for HUVEC), while gefitinib exhibited significant antitumor activity against HepG2 (IC_50_ = 6.42 μM). Except for **D04**, **D08**, **D09**, **D16,** and **D18**, hardly any the target compounds exhibited significant inhibition against the HeLa cells (IC_50_ > 100 μM).

Interestingly, the IC_50_ values against the HeLa cells of **D04**, **D08**, **D09**, **D16**, and **D18** were 0.42, 2.49, 14.05, 32.06, and 43.70 μM, respectively, where **D04** demonstrated the most potent antitumor activity and was superior in magnitude to gefitinib.

The preliminary structure–activity relationships analysis of the target compounds can be revealed as follows: (1) The active target compounds displayed excellent selectivity against the EGFR high-expressed HeLa cells, and hardly showed any activities against the EGFR low-expressed HepG2 cells and normal HL7702 and HUVEC cells; (2) When the substituents on the aniline ring were electron-withdrawing groups, the target compounds had nearly no antitumor activity against the HeLa cells except for **D09**, **D16**, and **D18**; and (3) When the substituents on the aniline ring were strong electron-donating groups such as **D04** (2-hydroxy) and **D08** (naphthalen-1-yl in the whole), they exhibited excellent antitumor activities against the HeLa cells; however, the target compounds **D05** and **D12** with middle electron-donating groups and the target compound **D15** with a weak electron-donating group showed nearly no activity against the HeLa cells. The target compound **D02** with 4-hydroxy had nearly no activity against the HeLa cells, which implies that the position of the electron-donating group may play a critical role in antitumor activity against HeLa cells. The antitumor results indicate that this is consistent with molecular docking.

#### 2.3.2. In Vitro EGFR Inhibitory Activity

To verify the effectiveness of the target compounds as potential EGFR inhibitors, the top three ranking target compounds in the anti-proliferative activities were evaluated for in vitro EGFR inhibitory activity studies, the results of which are shown in Table 3.

In general, the inhibitory activities against the EGFR kinase of the three target compounds were less than that of the positive controls gefitinib and osimertinib as an EGFR tyrosine kinase inhibitor that was approved by the FDA in 2017 [19]. However, the order of the inhibitory activities against the EGFR kinase of the three target compounds were greatly consistent with that of the anti-proliferative activities. This result showed that the target compounds with the benzo[4,5]imidazo[2,1-*b*]thiazole scaffold had an important contribution to the antitumor activity.

## 3. Experimental

### 3.1. Chemistry

Starting materials and solvents were purchased from common commercial suppliers and were used without further purification. Melting points were determined in open capillary tubes and uncorrected. Reaction progress was monitored by thin layer chromatography on silica gel sheets. The spots were visualized by ultraviolet (UV) light (254 nm). The IR spectra were obtained on a Bruker IFS55 spectrometer (Karlsruhe, Germany). NMR spectra were recorded on Bruker 400 MHz NMR spectrophotometers (Karlsruhe, Germany) using deuterated dimethyl sulfoxide (DMSO-d_6_) as the solvent and tetramethylsilane (TMS) as the internal standard. Mass spectra were measured with electrospray (ESI) on a Waters spectrometer (Milford, MA, USA). HRMS analyses were recorded on an Agilent Technologies 6530 Accurate-Mass Q-TOF Mass Spectrometer (Santa Clara, CA, USA).

The original figures of IR, 1H-NMR, 13C-NMR, ESI-MS, and HRMS of all the target compounds **D01**-**D18** as the Appendix A are available online.

#### 3.1.1. Preparation of 1*H*-Benzo[d]imidazole-2-thiol (2)

A stirred mixture of benzene-1,2-diamine (**1**, 10.8 g, 0.1 mol), CS_2_ (11.4 g, 0.15 mol), Na_2_CO_3_ (7.56 g, 0.07 mol), and 150 mL water was heated at 30 °C for 2 h, then at 60 °C for 2 h, and finally at 90 °C for 3 h. After the reaction was finished, the reaction solution was filtered and washed twice with hot water, so that a white solid of 14.25 g was obtained in a yield of 95% and subsequently used for the next reaction without further purification. m.p. 290–292 °C (see [20]): 303–305 °C); IR (KBr, cm^−1^): 3441, 3153, 3114, 2979, 2877, 1618, 1513, 1467, and 1215. ^1^H-NMR (400 MHz, DMSO-d_6_): δ 7.10–7.15 (m, 4H), 12.51 (s, 2H). ESI-MS (*m*/*z*): 151.1 ([M + H]^+^).

#### 3.1.2. Preparation of Ethyl 4-[(1*H*-Benzo[d]imidazol-2-yl)thio]-3-oxobutanoate (3)

A stirred mixture of compound **2** (1.50 g, 0.01 mol), ethyl 4-chloro-3-oxobutanoate (2.50 g, 0.015 mol), KI (0.01 g), Na_2_CO_3_, and 25 mL DMF was left at room temperature overnight, and after the reaction finished, we added 100 mL water, stirred for a quarter, and finally, the light-red product of 2.36 g was filtered and washed twice by water, dried at room temperature, and obtained in an 85% yield, before being used for the next reaction without further purification. m.p. 50–52 °C; IR (KBr, cm^−1^): 3437.4 (s), 2981.1 (m), 1730.2 (s), 1633.1 (s), 1478.4 (s), 1446.9 (s), 1325.1 (s), 1245.5 (s), 1135.1 (s), 741.6 (s). ^1^H-NMR (400 MHz, DMSO-d_6_): δ 1.18 (t, 3H, CH_3_, J = 7.1 Hz), 3.83 (s, 2H, CH_2_), 4.09 (q, 2H, CH_2_, J = 7.1 Hz), 4.42 (s, 2H, CH_2_), 7.12–7.15 (m, 2H, Ar–H), 7.44–7.46 (m, 1H, Ar–H), 7.55–7.57 (m, 1H, Ar–H), 12.60 (s, 1H, NH). ESI-MS (*m*/*z*): 279.1 ([M + H]^+^).

#### 3.1.3. Preparation of Ethyl 2-(Benzo[4,5]imidazo[2,1-*b*]thiazol-3-yl)acetate (4)

Concentrated sulfuric acid (14.70 g, 0.15 mol) was slowly added in a rounded-flask that contained compound **3** (8.16 g, 0.09 mol) dropwise until finished at 50 °C for 4 h, then cooled, and the reaction was adjusted neutrally by sodium hydroxide solution, before finally, the anticipated solid was filtered and washed twice with water, and a brown product of 2.13 g was obtained at a yield of 27% and used for the next reaction without further purification. m.p. 90–92 °C; IR (KBr, cm^−1^): 3096.8 (s), 2919.8 (m), 2858.7 (m), 1612.1 (m), 1592.0 (m), 1477.2 (s), 1459.2 (s), 1379.3 (s), 1269.6 (s), 1209.4 (s), 733.6 (s). ^1^H-NMR (400 MHz, DMSO-d_6_): δ 2.74 (d, 3H, CH_3_), 3.32 (s, 4H, CH_2_, CH_2_), 6.85–6.86 (m, 1H, =C–H), 7.23–7.27 (m, 1H, Ar–H, J = 7.6 Hz), 7.32–7.3 (m, 1H, Ar–H, J = 7.6 Hz), 7.67–7.69 (d, 1H, Ar–H, J = 8.0 Hz), 7.96–7.98 (d, 1H, Ar–H, J = 8.0 Hz). ESI-MS (*m*/*z*): 261.3 ([M + H]^+^).

#### 3.1.4. Preparation of 2-(Benzo[4,5]imidazo[2,1-*b*]thiazol-3-yl)acetic Acid (5)

A stirred mixture of compound **4** (1.31 g, 0.005 mol), NaOH (2 g, 0.05 mol), and 25 mL water was refluxed for 10 h, and then the diluted HCl (2 mol/L) solution was added in the finished reaction until neutral, mounds of white solid appeared, then was filtered and washed twice with water, thus obtaining a white product of 0.21 g in a yield of 18% and used for the next reaction without further purification. m.p.: 200–202 °C. IR (KBr, cm^−1^): 3428.0 (s), 2927.2 (m), 2853.5 (m), 1701.6 (s), 1461.2 (s), 1329.4 (s), 1234.3 (s), 1171.1 (m), 739.9 (s). ^1^H-NMR (400 MHz, DMSO-d_6_): δ 4.26 (s, 2H, CH_2_), 7.07 (s, 1H, =C–H), 7.07–7.26 (m, 1H, Ar–H, J = 7.7 Hz), 7.31–7.35 (m, 1H, Ar–H, J = 7.7 Hz), 7.67–7.69 (d, 1H, Ar–H, J = 8.4 Hz), 7.73–7.75 (d, 1H, Ar–H, J = 8.4 Hz), 12.52 (s, 1H, –COOH). ESI-MS (*m*/*z*): 233.2 ([M + H]^+^).

#### 3.1.5. General Procedure for the Preparation of 2-(Benzo[4,5]imidazo[2,1-*b*]thiazol-3-yl)acetamide Derivatives (D01 to D18)

A stirred mixture of compound **5** (0.23 g, 0.001 mol), EDCI (0.23 g, 0.0012 mol), HOBt (0.16 g, 0.0012 mol), aniline or aniline derivative (0.001 mol), and 20 mL DMF was reacted at room temperature overnight, and then 40 mL water was added and stirred for a quarter, before the crude was filtered and washed twice with water. The anticipated product was purified by column chromatography on silica gel using petroleum ether/ethyl acetate (2/1, volume ratio) as the eluent to afford the target compound.

2-(Benzo[4,5]imidazo[2,1-*b*]thiazol-3-yl)-*N*-phenylacetamide (**D01**): a white solid, yield 52%, m.p.: 123–125 °C. IR (KBr, cm^−1^): 3422.6 (s), 3057.4 (m), 2923.5 (s), 2853.3 (m), 1663.6 (s), 1544.2 (s), 1471.4 (s), 1453.2 (s), 1384.2 (s), 1254.8 (s), 1175.3 (m), 753.4 (s), 737.3 (s), 692.9 (s). ^1^H-NMR (400 MHz, DMSO-*d_6_*): δ 4.34 (s, 2H, CH_2_), 7.05–7.09 (m, 2H, =C–H, Ar–H), 7.19–7.23 (m, 1H, Ar–H), 7.30–7.34 (m, 3H, Ar–H), 7.59 (d, 2H, Ar–H, *J* = 7.8 Hz), 7.68 (d, 1H, Ar–H, *J* = 8.1 Hz), 7.85 (d, 1H, Ar–H, *J* = 8.1 Hz), 10.49 (s, 1H, N–H). ^13^C-NMR (101 MHz, DMSO-*d_6_*) δ166.13, 150.10, 149.29, 137.81, 129.12, 128.82 (2 × C), 127.55, 127.29, 124.69, 122.14, 120.95 (2 × C), 116.77, 112.52, 111.92, 35.19. ESI-MS (*m*/*z*): 308.1 ([M + H]^+^); HRMS (ESI) (*m*/*z*): [M – H]^−^ calcd for C_17_H_12_ClN_3_OS, 340.031684; found, 340.032253.

2-(Benzo[4,5]imidazo[2,1-*b*]thiazol-3-yl)-*N*-(4-hydroxyphenyl)acetamide (**D02**): a light-red solid, yield 61%, m.p.: 101–103 °C. IR (KBr, cm^−1^): 3450.5 (s), 2923.1 (s), 2852.3 (m), 1607.1 (s), 1554.7 (s), 1468.4 (s), 1382.8 (s), 1235.8 (s), 1152.3 (s), 861.9 (m), 785.6 (m), 739.4 (s), 689.3 (m). ^1^H-NMR (400 MHz, DMSO-*d_6_*): δ 4.27 (s, 2H, CH_2_), 6.70 (d, 2H, Ar–H, *J* = 8.8 Hz), 7.07 (s, 1H, =C–H), 7.19–7.23 (m, 1H, Ar–H), 7.29–7.31 (m, 1H, Ar–H), 7.34 (d, 2H, Ar–H, *J* = 8.8 Hz), 7.68 (d, 1H, Ar–H, *J* = 8.2 Hz), 7.85 (d, 1H, Ar–H, J = 8.2 Hz), 9.24 (s, 1H, –OH), 10.20 (s, 1H, N–H). ^13^C-NMR (101 MHz, DMSO-*d_6_*): δ 165.12, 155.44, 153.78, 141.76, 130.39, 129.64, 128.52, 124.86, 122.28, 121.27 (2 × C), 116.57, 115.22 (2 × C), 112.67, 111.99, 34.99. ESI-MS (*m*/*z*): 324.1 ([M + H]^+^); HRMS (ESI) (*m*/*z*): [M – H]^−^ calcd for C_17_H_12_ClN_3_OS, 322.065571; found, 322.066142.

2-(Benzo[4,5]imidazo[2,1-*b*]thiazol-3-yl)-*N*-(3-hydroxyphenyl)acetamide (**D03**): a white solid, yield 55%, m.p.: 111–113 °C. IR (KBr, cm^−1^): 3450.5 (s), 2923.1 (s), 2852.3 (m), 1607.1 (s), 1554.7 (s), 1468.4 (s), 1382.8 (s), 1235.8 (s), 1152.3 (s), 861.9 (m), 785.6 (m), 739.4 (s), 689.3 (m). ^1^H-NMR (400 MHz, DMSO-d_6_): δ 4.32 (s, 2H, CH_2_), 6.46–6.49 (m, 1H, Ar–H), 6.96–7.00 (m, 1H, Ar–H), 7.06–7.10 (m, 2H, Ar–H), 7.16 (s, 1H, =C–H), 7.19–7.23 (m, 1H, Ar–H), 7.30–7.33 (m, 1H, Ar–H), 7.70 (d, 1H, Ar–H, *J* = 8.0 Hz), 7.84 (d, 1H, Ar–H, *J* = 8.0 Hz), 7.95 (s, 1H, –OH), 10.39 (s, 1H, N–H). ^13^C-NMR (101 MHz, DMSO-d_6_): δ 166.14, 158.15, 155.92, 142.68, 140.24, 129.94, 129.79, 129.03, 125.10, 122.53, 117.15, 112.96, 112.16, 111.25, 110.51, 106.97, 35.65. ESI-MS (*m*/*z*): 324.2 ([M + H]^+^); HRMS(ESI) (*m*/*z*): [M – H]^−^ calcd for C_17_H_12_ClN_3_OS, 322.065571; found, 322.066120.

2-(Benzo[4,5]imidazo[2,1-*b*]thiazol-3-yl)-*N*-(2-hydroxyphenyl)acetamide (**D04**): a white solid, yield 49%, m.p.: 121–123 °C. IR (KBr, cm^−1^): 3442.8 (s), 3380.0 (s), 3094.9 (m), 2924.4 (m), 2851.3 (m), 1689.4 (s), 1528.1 (s), 1474.8 (s), 1449.9 (s), 1384.5 (s), 1279.0 (s), 1148.6 (m), 739.4 (s). ^1^H-NMR (400 MHz, DMSO-d_6_): δ 4.43 (s, 2H, CH_2_), 6.71–6.75 (m, 1H, Ar–H), 6.88–6.90(m, 1H, Ar–H), 6.92–6.96 (m, 1H, Ar–H), 7.09 (s, 1H, =C–H), 7.19–7.23 (m, 1H, Ar–H), 7.30–7.34 (m, 1H, Ar–H), 7.68 (d, 1H, Ar–H, *J* = 8.0 Hz), 7.75 (dd, 1H, Ar–H, *J*_1_ = 8.0 Hz, *J*_2_ = 1.2 Hz), 7.93 (d, 1H, Ar–H, *J* = 8.0 Hz), 9.70 (s, 1H, –OH), 9.90 (s, 1H, N–H).^13^C-NMR (101 MHz, DMSO-d_6_): δ 166.14, 158.15, 155.92, 142.68, 140.24, 129.94, 129.79, 129.03, 125.10, 122.53, 117.15, 112.96, 112.16, 111.25, 110.51, 106.97, 35.65. ESI-MS (*m*/*z*): 324.2 ([M + H]^+^); HRMS (ESI) (*m*/*z*): [M – H]^−^ calcd for C_17_H_12_ClN_3_OS, 322.065571; found, 322.066063.

*N*-(4-(Allyloxy)phenyl)-2-(benzo[4,5]imidazo[2,1-*b*]thiazol-3-yl)acetamide (**D05**): a white solid, yield 67%, m.p.: 196–198 °C. IR (KBr, cm^−1^): 3426.5 (s), 2924.7 (s), 1664.0 (s), 1611.4 (s), 1510.7 (s), 1472.6 (s), 1383.8 (s), 1241.1 (s), 1170.9 (m), 927.4 (m), 738.9 (s). ^1^H-NMR (400 MHz, DMSO-d_6_): δ 4.30 (s, 2H, CH_2_), 4.50–4.53(m, 2H, CH_2_), 5.24 (dd, 1H, –C=C–H, *J*_1_ = 10.5 Hz, *J*_2_ = 1.5 Hz), 5.37 (dd, 1H, –C=C–H, *J*_1_ = 17.3 Hz, *J*_2_ = 1.5 Hz), 5.97–6.07 (m, 1H, –C=C–H), 6.91 (d, 2H, Ar–H, *J* = 9.0 Hz), 7.09 (s, 1H, =C–H), 7.20–7.24 (m, 1H, Ar–H), 7.30–7.34 (m, 1H, Ar–H), 7.47 (d, 2H, Ar–H, J = 9.0 Hz), 7.70 (d, 1H, Ar–H, *J* = 8.1 Hz), 7.85 (d, 1H, Ar–H, *J* = 8.1 Hz), 10.30 (s, 1H, N–H). ^13^C-NMR (101 MHz, DMSO-d_6_): δ 165.78, 155.84, 154.87, 142.21, 134.27, 132.49, 129.93, 128.93, 125.23, 122.65, 121.38 (2 × C), 117.81, 116.98, 115.24 (2 × C), 113.04, 112.42, 68.79, 35.44. ESI-MS (*m*/*z*): 364.0 ([M + H]^+^); HRMS (ESI) (*m*/*z*): [M – H]^−^ calcd for C_17_H_12_ClN_3_OS, 362.096871; found, 362.097386.

2-(Benzo[4,5]imidazo[2,1-*b*]thiazol-3-yl)-*N*-(3-nitrophenyl)acetamide (**D06**): a white solid, yield 29%, m.p.: 132–134 °C. IR (KBr, cm^−1^): 3438.3 (s), 2925.7 (m), 1617.1 (s), 1529.0 (s), 1475.9 (s), 1383.9 (s), 1348.7 (s), 737.9 (s). ^1^H-NMR (400 MHz, DMSO-d_6_): δ 4.41 (s, 2H, CH_2_), 7.13 (s, 1H, =C–H), 7.19–7.23 (m, 1H, Ar–H), 7.30–7.33 (m, 1H, Ar–H), 7.62–7.66 (m, 1H, Ar–H), 7.69 (d, 1H, Ar–H, *J* = 8.1 Hz), 7.82 (d, 1H, Ar–H, *J* = 8.1 Hz), 7.92–7.96 (m, 2H, Ar–H), 8.59–8.60 (m, 1H, Ar–H), 10.95 (s, 1H, N-H). ^13^C-NMR (101 MHz, DMSO-d_6_): δ 167.17, 155.45, 148.42, 140.42, 140.15, 130.75, 129.66, 128.46, 125.87, 125.73, 123.32, 118.62, 116.34, 113.88, 113.80, 113.44, 35.51. ESI-MS (*m*/*z*): 353.0 ([M + H]^+^); HRMS (ESI) (*m*/*z*): [M – H]^−^ calcd for C_17_H_12_ClN_3_OS, 351.055735; found, 351.056060.

2-(Benzo[4,5]imidazo[2,1-*b*]thiazol-3-yl)-*N*-(4-nitrophenyl)acetamide (**D07**): a light-yellowish solid, yield 34%, m.p.: 99–101 °C. IR (KBr, cm^-1^): 3427.2 (s), 2922.8 (s), 2852.5 (m), 1620.0 (s), 1563.5 (s), 1502.5 (s), 1466.8 (s), 1383.9 (s), 1334.8 (s), 1270.6 (s), 1152.5 (m), 854.0 (m), 731.1 (m). ^1^H-NMR (400 MHz, DMSO-d_6_): δ 4.44 (s, 2H, CH_2_), 7.13 (s, 1H, =C–H), 7.19–7.22 (m, 1H, Ar–H), 7.29–7.33 (m, 1H, Ar–H), 7.69 (d, 1H, Ar–H, *J* = 8.2 Hz), 7.81 (d, 1H, Ar–H, *J* = 8.2 Hz), 7.85 (d, 2H, Ar–H, *J* = 9.0 Hz), 8.24 (d, 2H, Ar–H, *J* = 9.0 Hz), 11.15 (s, 1H, N–H). ^13^C-NMR (101 MHz, DMSO-d_6_): δ 167.12, 155.71, 145.00, 143.39, 142.55, 128.85, 128.55, 125.12 (2 x C), 124.38, 121.86, 119.09 (2 x C), 117.12, 112.32, 111.58, 35.44. ESI-MS (*m*/*z*): 353.1 ([M +H ]^+^); HRMS (ESI) (*m*/*z*): [M – H]^−^ calcd for C_17_H_12_ClN_3_OS, 351.055735; found, 351.056040.

2-(Benzo[4,5]imidazo[2,1-*b*]thiazol-3-yl)-*N*-(naphthalen-1-yl)acetamide (**D08**): a white solid, yield 69%, m.p.: 136–138 °C. IR (KBr, cm^−1^): 3437.7 (s), 2924.0 (s), 2852.8 (s), 1649.9 (s), 1543.4 (s), 1477.5 (s), 1384.2 (s), 1262.3 (s), 767.3 (m), 738.5 (s). ^1^H-NMR (400 MHz, DMSO-d_6_): δ 4.54 (s, 2H, CH_2_), 7.16 (s, 1H, =C–H), 7.24–7.27 (m, 1H, Ar–H), 7.32–7.36 (m, 1H, Ar–H), 7.47–7.51 (m, 1H, Ar–H), 7.54–7.61 (m, 2H, Ar–H), 7.66–7.72 (m, 2H, Ar–H), 7.78 (d, 1H, Ar–H, *J* = 8.1 Hz), 7.95–7.97 (m, 1H, Ar–H), 8.03 (d, 1H, Ar–H, *J* = 8.1 Hz), 8.20 (d, 1H, Ar–H, *J* = 8.0 Hz), 10.49 (s, 1H, N–H). ^13^C-NMR (101 MHz, DMSO-d_6_): δ 166.77, 155.18, 140.11, 133.80, 133.18, 129.83, 128.24, 128.16, 127.88, 126.22, 126.03, 125.73, 125.61, 125.41, 122.89, 122.73, 121.91, 116.08, 113.13, 113.02, 34.84. ESI-MS (*m*/*z*): 358.0 ([M + H]^+^); HRMS (ESI) (*m*/*z*): [M – H]^−^ calcd for C_17_H_12_ClN_3_OS, 356.086307; found, 356.087045.

2-(Benzo[4,5]imidazo[2,1-*b*]thiazol-3-yl)-*N*-(4-chlorophenyl)acetamide (**D09**): a white solid, yield 64%, m.p.: 151–153 °C. IR (KBr, cm^−1^): 3436.8 (s), 3059.1 (m), 2924.5 (m), 2853.7 (m), 1659.3 (s), 1509.5 (s), 1477.4 (s), 1384.2 (s), 1255.0 (s), 1155.8 (m), 1090.8 (m), 825.3 (s), 739.1 (s). ^1^H-NMR (400 MHz, DMSO-d_6_): δ 4.35 (s, 2H, CH_2_), 7.09 (s, 1H, =C–H), 7.19–7.23 (m, 1H, Ar–H), 7.29–7.33 (m, 1H, Ar–H), 7.38 (d, 2H, Ar–H, *J* = 8.8 Hz), 7.61 (d, 2H, Ar–H, *J* = 8.8 Hz), 7.68 (d, 1H, Ar–H, *J* = 8.0 Hz), 7.82 (d, 1H, Ar–H, *J* = 8.0 Hz), 10.61 (s, 1H, NH). ^13^C-NMR (101 MHz, DMSO-d_6_): δ 166.15, 156.56, 139.05, 128.84 (2 × C), 123.59, 123.06, 123.00, 120.53, 120.49, 119.39 (2 × C), 118.43, 118.39, 111.64, 111.48, 35.47. ESI-MS (*m*/*z*): 342.1 ([M + H]^+^); HRMS (ESI) (*m*/*z*): [M – H]^−^ calcd for C_17_H_12_ClN_3_OS, 306.070657; found, 306.070300.

2-(Benzo[4,5]imidazo[2,1-*b*]thiazol-3-yl)-*N*-(4-chloro-3-(trifluoromethyl)phenyl)acetamide (**D10**): a white solid, yield 45%, m.p.: 153–155 °C. IR (KBr, cm^-1^): 3437.1 (s), 2925.3 (m), 2854.1 (m), 1628.8 (s), 1547.5 (m), 1479.5 (s), 1384.1 (s), 1262.4 (m), 1131.5 (s), 1033.7 (m), 896.4 (m), 830.8 (m), 740.5 (m). ^1^H-NMR (400 MHz, DMSO-d_6_): δ 4.40 (s, 2H, CH_2_), 7.13 (s, 1H, =C–H), 7.19–7.23 (m, 1H, Ar–H), 7.30–7.34 (m, 1H, Ar–H), 7.67–7.71 (m, 2H, Ar–H), 7.81–7.86 (m, 2H, Ar–H), 8.18 (d, 1H, Ar–H, *J* = 2.0 Hz), 11.01 (s, 1H, NH). ^13^C-NMR (101 MHz, DMSO-d_6_): δ 166.93, 155.93, 138.35, 132.44, 129.04, 128.57, 124.84, 124.35, 124.29, 122.31, 121.84, 118.08, 118.03, 117.37, 112.54, 112.27, 111.38, 35.37. ESI-MS (*m*/*z*): 410.0 ([M + H]^+^); HRMS (ESI) (*m*/*z*): [M – H]^−^ calcd for C_17_H_12_ClN_3_OS, 408.019069; found, 408.019311.

2-(Benzo[4,5]imidazo[2,1-*b*]thiazol-3-yl)-*N*-(4-bromo-3-fluorophenyl)acetamide (**D11**): a white solid, yield 42%, m.p.: 151–153 °C. IR (KBr, cm^−1^): 3439.6 (s), 2924.4 (s), 2854.2 (m), 1692.0 (s), 1548.9 (s), 1494.7 (s), 1469.4 (s), 1385.9 (s), 1258.2 (s), 1042.5 (m), 875.8 (m), 821.3 (m), 737.3 (s). ^1^H-NMR (400 MHz, DMSO-d_6_): δ 4.39 (s, 2H, CH_2_), 7.11 (s, 1H, =C–H), 7.19–7.22 (m, 1H, Ar–H), 7.29–7.36 (m, 2H, Ar–H), 7.63–7.69 (m, 2H, Ar–H), 7.74 (d, 1H, Ar–H, *J* = 8.1 Hz), 7.83 (d, 1H, Ar–H, *J* = 8.1 Hz), 11.04 (s, 1H, NH). ^13^C-NMR (101 MHz, DMSO-d_6_): δ 167.14, 156.93, 148.31, 140.45, 140.35, 133.97, 130.25, 128.31, 123.40, 120.91, 118.93, 117.08, 111.84, 109.43, 107.89, 107.62, 35.93. ESI-MS (*m*/*z*): 401.6 ([M − H]^−^); HRMS (ESI) (*m*/*z*): [M – H]^−^ calcd for C_17_H_12_ClN_3_OS, 401.971747; found, 401.972353.

2-(Benzo[4,5]imidazo[2,1-*b*]thiazol-3-yl)-*N*-(4-(benzyloxy)phenyl)acetamide (**D12**): a white solid, yield 61%, m.p.: 135–137 °C. IR (KBr, cm^−1^): 3415.6 (s), 3061.8 (m), 2923.9 (s), 2854.1 (m), 1655.9 (s), 1510.1 (s), 1471.3 (s), 1454.2 (s), 1384.0 (s), 1239.5 (s), 1169.1 (s), 1023.4 (m), 829.3 (s), 738.2 (s), 698.0 (s). ^1^H-NMR (400 MHz, DMSO-d_6_): δ 4.30 (s, 2H, CH_2_), 5.06 (s, 2H, CH_2_), 5.96–6.98 (m, 2H, Ar–H), 7.08 (s, 1H, =C–H), 7.19–7.23(m, 1H, Ar–H), 7.29–7.33 (m, 2H, Ar–H), 7.36–7.44 (m, 4H, Ar–H), 7.47–7.50 (m, 2H, Ar–H), 7.68 (d, 1H, Ar–H, *J* = 8.1 Hz), 7.85 (d, 1H, Ar–H, *J* = 8.1 Hz), 10.36 (s, 1H, NH). ^13^C-NMR (101 MHz, DMSO-d_6_): δ 165.64, 156.00, 154.83, 143.96, 137.40, 132.35, 129.40, 129.15, 128.67 (2 × C), 128.05, 127.91 (2 × C), 124.46, 121.90, 121.19 (2 × C), 117.39, 115.22 (2 × C), 112.48, 111.19, 69.62, 35.33. ESI-MS (*m*/*z*): 414.2 ([M + H]^+^); HRMS (ESI) (*m*/*z*): [M – H]^−^ calcd for C_17_H_12_ClN_3_OS, 412.112521; found, 412.113094.

2-(Benzo[4,5]imidazo[2,1-*b*]thiazol-3-yl)-*N*-(3-bromo-4-fluorophenyl)acetamide (**D13**): a white solid, yield 29%, m.p.: 218–220 °C. IR (KBr, cm^−1^): 3439.6 (s), 2924.4 (s), 2854.2 (m), 1692.0 (s), 1548.9 (s), 1494.7 (s), 1469.4 (s), 1385.9 (s), 1258.2 (s), 1042.5 (m), 875.8 (m), 821.3 (m), 737.3 (s). ^1^H-NMR (400 MHz, DMSO-d_6_): δ 4.41 (s, 2H, CH2), 7.25 (s, 1H, =C–H), 7.29–7.33 (m, 1H, Ar–H), 7.34–7.38 (m, 1H, Ar-H), 7.40–7.42 (m, 1H, Ar–H), 7.52–7.56 (m, 1H, Ar–H), 7.76 (d, 1H, Ar–H, *J* = 8.2 Hz), 7.89 (d, 1H, Ar–H, *J* = 8.2 Hz), 8.02 (dd, 1H, Ar–H, *J*_1_ = 6.4 Hz, *J*_2_ = 2.6 Hz), 10.82 (s, 1H, NH). ^13^C-NMR (101 MHz, DMSO-d_6_): δ 166.23, 155.20, 136.41, 136.38, 129.29, 128.23, 125.26, 123.55, 122.71, 120.40, 120.33, 117.03, 116.80, 116.19, 112.99, 112.90, 35.03. ESI-MS (*m*/*z*): 401.6 ([M – H]^−^); HRMS (ESI) (*m*/*z*): [M – H]^−^ calcd for C_17_H_12_ClN_3_OS, 401.971747; found, 401.971380.

2-(Benzo[4,5]imidazo[2,1-*b*]thiazol-3-yl)-*N*-(4-bromophenyl)acetamide (**D14**): a white solid, yield 23%, m.p.: 107–109 °C. IR (KBr, cm^−1^): 3449.2 (s), 2924.6 (m), 2854.0 (m), 1659.4 (s), 1546.1 (s), 1471.6 (s), 1384.2 (s), 1256.3 (m), 1119.4 (m), 1072.5 (m), 821.8 (m), 739.4 (m). ^1^H-NMR (400 MHz, DMSO-d_6_): δ 4.35 (s, 2H, CH_2_), 7.09 (s, 1H, =C–H), 7.18–7.23 (m, 1H, Ar–H), 7.29–7.33 (m, 1H, Ar–H), 7.49–7.51 (m, 2H, Ar–H), 7.56–7.58 (m, 2H, Ar–H), 7.68 (d, 1H, Ar–H, *J* = 8.1 Hz), 7.82 (d, 1H, Ar–H, *J* = 8.1 Hz), 10.63 (s, 1H, NH). ^13^C-NMR (101 MHz, DMSO-d_6_): δ 166.60, 156.21, 144.30, 138.65, 132.12 (2 x C), 129.37, 129.25, 124.61, 122.06, 121.73 (2 × C), 117.65, 115.71, 112.63, 111.46, 35.70. ESI-MS (*m*/*z*): 385.9 ([M + H]^+^); HRMS (ESI) (*m*/*z*): [M – H]^−^ calcd for C_17_H_12_ClN_3_OS, 383.981169; found, 383.981719.

2-(Benzo[4,5]imidazo[2,1-*b*]thiazol-3-yl)-*N*-(2-methylphenyl)acetamide (**D15**): a white solid, yield 42%, m.p.: 83–85 °C. IR (KBr, cm^−1^): 3425.5 (s), 2923.6 (s), 2852.6 (s), 1656.1 (s), 1528.8 (m), 1467.1 (s), 1384.1 (s), 1263.1 (m), 742.2 (s). ^1^H-NMR (400 MHz, DMSO-d_6_): δ 2.25 (s, 3H, CH_3_), 4.38 (s, 2H, CH_2_), 7.21–7.27 (m, 3H, Ar–H), 7.31–7.38 (m, 3H, Ar–H), 7.53 (s, 1H, =C–H), 7.70 (d, 1H, Ar–H, *J* = 8.0 Hz), 7.94 (d, 1H, Ar–H, *J* = 8.0 Hz), 9.85 (s, 1H, NH). ^13^C-NMR (101 MHz, DMSO-d_6_): δ 166.58, 157.00, 148.30, 136.40, 132.28, 130.88, 128.96, 126.50, 125.97, 125.53, 123.44, 120.77, 118.91, 111.96, 109.24, 105.73, 35.41, 18.35. ESI-MS (*m*/*z*): 322.0 ([M + H]^+^); HRMS (ESI) (*m*/*z*): [M – H]^−^ calcd for C_17_H_12_ClN_3_OS, 320.086307; found, 320.086912.

2-(Benzo[4,5]imidazo[2,1-*b*]thiazol-3-yl)-*N*-(3,5-difluorophenyl)acetamide (**D16**): a white solid, yield 21%, m.p.: 101–103 °C. IR (KBr, cm^−1^): 3430.4 (s), 2924.9 (s), 2854.0 (s), 1624.3 (s), 1466.7 (s), 1383.9 (s), 1268.1 (m), 1116.3 (m), 743.5 (s). ^1^H-NMR (400 MHz, DMSO-d_6_): δ 4.38 (s, 2H, CH_2_), 7.11–7.13 (m, 3H, =C–H, Ar–H), 7.20–7.24 (m, 2H, Ar–H), 7.53 (s, 1H, Ar–H), 7.69 (d, 1H, Ar–H, J = 8.4 Hz), 7.80 (d, 1H, Ar–H, J = 8.4 Hz), 10.92 (s, 1H, NH). ^13^C-NMR (101 MHz, DMSO-d_6_): δ 166.71, 163.79, 163.63, 161.36, 161.22, 155.22, 140.69, 129.05, 128.25, 125.19, 122.65, 116.25, 112.99, 112.84, 102.36, 102.07, 35.17. ESI-MS (*m*/*z*): 341.7 ([M – H]^−^); HRMS (ESI) (*m*/*z*): [M – H]^−^ calcd for C_17_H_12_ClN_3_OS, 342.051813; found, 342.052163.

2-(Benzo[4,5]imidazo[2,1-*b*]thiazol-3-yl)-*N*-(2,4-dichlorophenyl)acetamide (**D17**): a white solid, yield 21%, m.p.: 206–208 °C. IR (KBr, cm^−1^): 3439.4 (s), 2923.8 (m), 2853.8 (m), 1674.4 (s), 1527.9 (s), 1467.4 (s), 1384.1 (s), 1262.8 (m), 1101.1 (m), 863.2 (m), 820.9 (m), 738.3 (m). ^1^H-NMR (400 MHz, DMSO-d_6_): δ 4.44 (s, 2H, CH_2_), 7.12 (s, 1H, =C–H), 7.22–7.25 (m, 1H, Ar–H), 7.31–7.35 (m, 1H, Ar–H), 7.41 (dd, 1H, Ar–H, *J*_1_ = 8.7 Hz, *J*_2_ = 2.0 Hz), 7.69 (d, 1H, Ar–H, *J* = 8.7 Hz), 7.91 (d, 1H, Ar–H, *J* = 8.0 Hz), 10.20 (s, 1H, NH). ^13^C-NMR (101 MHz, DMSO-d_6_): δ 166.65, 155.65, 143.04, 133.73, 130.02, 129.13, 128.83, 128.72, 127.85, 127.74, 127.52, 124.51, 121.88, 117.01, 112.38, 111.69, 34.74. ESI-MS (*m*/*z*): 373.8 ([M − H]^−^); HRMS (ESI) (*m*/*z*): [M – H]^−^ calcd for C_17_H_12_ClN_3_OS, 373.992712; found, 373.992947.

2-(Benzo[4,5]imidazo[2,1-*b*]thiazol-3-yl)-*N*-(4-(trifluoromethyl)phenyl)acetamide (**D18**): a white solid, yield 19%, m.p.: 65–67 °C. IR (KBr, cm^−1^): 3433.0 (s), 2926.1 (m), 2854.7 (m), 1606.4 (s), 1524.9 (s), 1471.3 (s), 1410.4 (s), 1384.1 (s), 1322.5 (s), 1114.3 (s), 1065.9 (s), 843.4 (m), 740.3 (m). ^1^H-NMR (400 MHz, DMSO-d_6_): δ 4.33–4.40 (m, 2H, CH_2_), 7.11 (s, 1H, =C–H), 7.19–7.23 (m, 1H, Ar–H), 7.29–7.33 (m, 1H, Ar–H), 7.43 (d, 1H, Ar–H, *J* = 8.0 Hz), 7.55 (d, 1H, Ar–H, *J* = 8.5 Hz), 7.68–7.71 (m, 2H, Ar–H), 7.80–7.83 (m, 2H, Ar–H), 10.85 (s, 1H, NH). ^13^C-NMR (101 MHz, DMSO-d_6_): δ 166.75, 155.81, 142.47, 133.90, 128.97. ESI-MS (*m*/*z*): 373.8 ([M – H]^−^); HRMS (ESI) (*m*/*z*): [M – H]^−^ calcd for C_17_H_12_ClN_3_OS, 374.058041; found, 374.058608.

### 3.2. In Vitro Antitumor Activity Assay

The human cancer cell lines (HeLa and HepG2) and the human normal ones (HL7702 and HUVEC) were chosen to evaluate the antitumor activity and cell toxicity of target compounds in vitro by the MTT assay [21,22,23,24]. HeLa, HepG2, and HUVEC cells were cultured in a Dulbecco’s modified Eagle’s medium (DMEM) supplemented with 10% fetal bovine serum, and HL7702 cells were cultured in a DMEM supplemented with 10% fetal bovine serum. Cells were grown to 80–90% confluence and subjected to no more than 20 cell passages and were left to acclimatize for 24 h before any treatments. The cells were seeded into 96-well plates (8 * 104 cells/mL) and incubated at 37 °C in a humidified atmosphere containing 5% CO_2_ overnight. After 24 h, the cancer cells were treated with the target compounds at 100, 50, 25, 12.5, 6.25, 3.13, and 1.56 μmol/L; the human normal cells HL7702 were treated with 200 and 100 μmol/L of the target ones; and the HUVEC were treated with 100 and 50 μmol/L of the target compounds. After 24 h of incubation, MTT (5 mg/mL) was subsequently added and incubated for 4 h. Next, the culture medium was cleared, and the crystals were dissolved in DMSO. The optical densities (OD) at 490 nm were measured on a universal microplate reader (Bio-Tek instruments, Inc., EL × 800). Inhibitory effects were expressed as the percentage of IC_50_ value.

### 3.3. In Vitro Enzymatic Activity Assay

The EGFR kinase assays of the target compounds were performed by the intrinsic ATPase activity luminescence assay [25]. All the enzymatic reactions were conducted at 30 °C for 40 minutes. The 50 µL reaction mixture contained 40 mM Tris, pH 7.4, 10 mM MgCl2, 0.1 mg/mL BSA, 1 mM DTT, 10 µM adenosine triphosphate (ATP), 0.2 ug/mL PI3 Kinase, and 100 µM lipid substrate. The compounds were diluted in 10% DMSO and 5 µL of the dilution was added to a 50 µL reaction so that the final concentration of DMSO was 1% in all the reactions. The assay was performed using the Kinase-Glo Plus luminescence kinase assay kit, which measures the kinase activity by quantifying the amount of ATP remaining in solution following a kinase reaction. The luminescent signal from the assay was correlated with the amount of ATP present and was inversely correlated with the amount of kinase activity. The IC_50_ values were calculated using nonlinear regression with a normalized dose−response fit using the Prism GraphPad software (version 6.0, GraphPad Software, San Diego, CA, USA).

## 4. Conclusions

A novel series of 2-(benzo[4,5]imidazo[2,1-*b*]thiazol-3-yl)acetamide derivatives was synthesized and their antitumor activities against the cancer cell lines HeLa and HepG2, which are abundant and low in the expression of EGFR, respectively, and the human normal cell lines (HL7702 and HUVEC) were evaluated by the MTT colorimetric assay. The biological screening results showed that some of the target compounds had potent antitumor activities against HeLa cells, weak antitumor activities against HepG2 cells, and low toxicities against human normal cells (HL7702 and HUVEC); moreover, a preliminary structure–activity relationship was clearly discerned. Meanwhile, the EGFR inhibitory activities in vitro were greatly consistent with the anti-proliferative activities. In summary, it is heralded that the benzo[4,5]imidazo[2,1-*b*]thiazole derivatives can be potential candidates as novel potential EGFR inhibitors in further research.

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
