# Peer review of "Synthesis, Characterization, and Biological Activity of a Novel Series of Benzo[4,5]imidazo[2,1-b]thiazole Derivatives as Potential Epidermal Growth Factor Receptor Inhibitors"

_molecules, 2019, doi:10.3390/molecules24040682_

Round 1

Reviewer 1 Report

The paper reports the synthesis of a series of Benzo[4,5]imidazo[2,1-b]thiazole drivatives and their investigation as potential epidermal growth factor receptor inhibitors.

The consept and methodology are well described.

However there are some major issues:

-The novel products were not characterised adequately. In particular, 13C-NMR data and elemental analysis are missing. Those are necessary for full structural determination of the new compounds and proof of their purity.

-The english needs improving in the paper. There are many grammatical errors.

I suggest that the paper is accepted after those two issues are addressed.

Author Response

The paper reports the synthesis of a series of Benzo[4,5]imidazo[2,1-b]thiazole derivatives and their investigation as potential epidermal growth factor receptor inhibitors.

The concept and methodology are well described.

However there are some major issues:

-The novel products were not characterised adequately. In particular, 13C-NMR data and elemental analysis are missing. Those are necessary for full structural determination of the new compounds and proof of their purity.

RESPONSE: According to the reviewer’s suggestions, we have added the 13C-NMR data and high resolution mass spectra data for full structural determination of the target compounds.  The purities of the target compounds and intermediates were controlled by the thin layer chromatography with different developing solvent system.

-The English needs improving in the paper. There are many grammatical errors.

RESPONSE: According to the reviewer’s suggestions, we have revised the language errors.

Reviewer 2 Report

The aim of the work is poor.

Many compounds described (for example the compounds 1, 2, 3, 4, 5) are known and some compounds D are described in a patent.

Very often the language does not allow a clear understanding.

The manuscript has mistakes.

The brute formula and the molecular weight of compounds are not reported, thus the adequate control of the described NMR and ESI-MS is difficult.

The relationship between virtual screening and activity is not clear and even docking results do not provide interesting information.

The activity is weak.

The manuscript reports that table 2 also indicates inhibitory ratios (%) but these data are not present.

The conclusions are poor.  

Author Response

The aim of the work is poor.

RESPONSE: The aim of our manuscript is mainly to discover new potential EGFR inhibitors. According the pharmacological data, some target compounds have shown significantly antitumor activity against the EGFR high-expressed human cell line HeLa, the inhibitory activities against EGFR kinase in vitro of three target compounds were greatly consistent with the anti-proliferative activities, and all the target compounds have shown hardly any antitumor activity against the EGFR low-expressed human cell line HepG2, and nearly no cellular toxicity against human normal cell lines HL7702 and HUVEC. In summary, We have reached the preliminary aim to discover the new antitumor agents with low toxicities.

Many compounds described (for example the compounds 1, 2, 3, 4, 5) are known and some compounds D are described in a patent.

RESPONSE: All the target compounds are new compounds, and are patented by the authors and their affiliation. The compounds 1, 2, 3, 4, 5 are only intermediates. Therefore, our manuscript is of the novelty and originality.

Very often the language does not allow a clear understanding. The manuscript has mistakes.

RESPONSE: According to the reviewer’s suggestions, we have revised the language errors.

The brute formula and the molecular weight of compounds are not reported, thus the adequate control of the described NMR and ESI-MS is difficult.

RESPONSE: According to the reviewer’s suggestions, we have added the 13C-NMR data and high resolution mass spectra data for the structural determination of the target compounds.

The relationship between virtual screening and activity is not clear and even docking results do not provide interesting information.

RESPONSE: The molecular docking results have showed that the binding energy and the H-bond energy of the target compounds reveal as good as the lead compound’s, and some target compounds are very similar interaction to EGFR with gefitinib, which can be inferred that the design of the target compound is reasonable and acceptable. According the pharmacological data, some target compounds have shown significantly antitumor activity against the EGFR high-expressed human cell line HeLa, the inhibitory activities against EGFR kinase in vitro of three target compounds were greatly consistent with the anti-proliferative activities. In conclusion, the relationship between virtual screening and activity is consistent.

The activity is weak. The manuscript reports that table 2 also indicates inhibitory ratios (%) but these data are not present.

RESPONSE: The activities of some target compounds are weak, but some target compounds have shown significantly antitumor activity against the EGFR high-expressed human cell line HeLa, the inhibitory activities against EGFR kinase in vitro of three target compounds were greatly consistent with the anti-proliferative activities. Especially, these active compounds have not antitumor activity against the EGFR low-expressed human cell line HepG2, and nearly no cellular toxicity against human normal cell lines HL7702 and HUVEC.

According to the reviewer’s suggestions, the extensive pharmacological test for the viable antitumor candidates and optimization of the target compounds should be studied in the near future in our laboratory.

The conclusions are poor.  

RESPONSE: According to the reviewer’s suggestions, we have revised the conclusions section.

Reviewer 3 Report

Editoral:

1) Figure 2 and 4 must be improved (better resolution)

2) Use "Figure" or "Fig" not both

Metodology

1) In the experimental section, the purity of the obtained compounds was not given, please add Elemental analysis, HLPC or HRMS to the characteristics.

2) Please add 13C NMR to the characteristics

3) There is no supporting files, please add this file with all NMR and IR spectra

Author Response

1) Figure 2 and 4 must be improved (better resolution)

2) Use "Figure" or "Fig" not both

RESPONSE: According to the reviewer’s suggestions, we have revised the Figure 2 and 4, and only use Figure.

Metodology

1) In the experimental section, the purity of the obtained compounds was not given, please add Elemental analysis, HLPC or HRMS to the characteristics.

2) Please add 13C NMR to the characteristics

3) There is no supporting files, please add this file with all NMR and IR spectra

RESPONSE: According to the reviewer’s suggestions, we have added the HRMS data and 13C-NMR data for full structural determination of the target compounds, and we provide the supporting files including all NMR, IR, MS, HRMS data.

Round 2

Reviewer 1 Report

The paper can now be accepted.

Author Response

The manuscript is edited and revised by MDPI, and text has been checked for correct use of grammar and common technical terms.

Reviewer 2 Report

The authors confirm that intermediates 1, 2, 3, 4 and 5 are known but the references are not mentioned as well as for some compounds D.

Table 2 also indicates inhibitory ratios (%) but these data are not present.

An extensive editing of Inglese language is required.

I'll give just a few examples, by indicating the number relative to the line, where the changes can be made: 16 (there are many "with"), 19 (add for), 32-34 (unclear), 50 (add "and"), 95 (add "in"), 123 (unclear).

Author Response

The authors confirm that intermediates 1, 2, 3, 4 and 5 are known but the references are not mentioned as well as for some compounds D.

RESPONSE: All the target compounds and the intermediates 3, 4, 5 are new compounds, and are patented by the authors. The compound 1 is only raw material. The intermediates 2 is a compound reported by Zhivotova T S, and the citation has marked.

Table 2 also indicates inhibitory ratios (%) but these data are not present.

RESPONSE: According to the reviewer’s suggestions, we have revised the table 2.

An extensive editing of Inglese language is required.

I'll give just a few examples, by indicating the number relative to the line, where the changes can be made: 16 (there are many "with"), 19 (add for), 32-34 (unclear), 50 (add "and"), 95 (add "in"), 123 (unclear).

RESPONSE: According to the reviewer’s suggestions, our manuscript is edited and revised by MDPI, and text has been checked for correct use of grammar and common technical terms.

Reviewer 3 Report

Thank you for adding the missing data and the supplement file

Author Response

(The authors gave the same response as above.)
